# The UPR Maintains Proteostasis and the Viability and Function of Hippocampal Neurons in Adult Mice

**DOI:** 10.3390/ijms241411542

**Published:** 2023-07-16

**Authors:** Pingting Liu, Md Razaul Karim, Ana Covelo, Yuan Yue, Michael K. Lee, Wensheng Lin

**Affiliations:** 1Department of Neuroscience, University of Minnesota, Minneapolis, MN 55455, USA; liup@stanford.edu (P.L.); karimr@vanquabio.com (M.R.K.); ana.covelo@inserm.fr (A.C.); yuan.yue@alphabiopharma.com (Y.Y.); mklee@umn.edu (M.K.L.); 2Institute for Translational Neuroscience, University of Minnesota, 2101 6th Street SE, WMBB4-140, Minneapolis, MN 55455, USA

**Keywords:** UPR, neuron, proteostasis, autophagy, lysosome, tau

## Abstract

The unfolded protein response (UPR), which comprises three branches: PERK, ATF6α, and IRE1, is a major mechanism for maintaining cellular proteostasis. Many studies show that the UPR is a major player in regulating neuron viability and function in various neurodegenerative diseases; however, its role in neurodegeneration is highly controversial. Moreover, while evidence suggests activation of the UPR in neurons under normal conditions, deficiency of individual branches of the UPR has no major effect on brain neurons in animals. It remains unclear whether or how the UPR participates in regulating neuronal proteostasis under normal and disease conditions. To determine the physiological role of the UPR in neurons, we generated mice with double deletion of PERK and ATF6α in neurons. We found that inactivation of PERK and ATF6α in neurons caused lysosomal dysfunction (as evidenced by decreased expression of the V0a1 subunit of v-ATPase and decreased activation of cathepsin D), impairment of autophagic flux (as evidenced by increased ratio of LC3-II/LC3-I and increased p62 level), and accumulation of p-tau and Aβ42 in the hippocampus, and led to impairment of spatial memory, impairment of hippocampal LTP, and hippocampal degeneration in adult mice. These results suggest that the UPR is required for maintaining neuronal proteostasis (particularly tau and Aβ homeostasis) and the viability and function of neurons in the hippocampus of adult mice.

## 1. Introduction

Accumulation of unfolded/misfolded proteins in the endoplasmic reticulum (ER) leads to ER stress and activation of the unfolded protein response (UPR), which comprises three parallel branches: pancreatic ER kinase (PERK), activating transcription factor 6α (ATF6α), inositol requiring enzyme 1 (IRE1) [1,2,3]. PERK activation phosphorylates eukaryotic translation initiation factor 2α (eIF2α), which inhibits protein translation but stimulates the expression of genes related to autophagy and genes related to ER-associated degradation (ERAD) by inducing the transcription factor ATF4. ATF6α undergoes cleavage, and the cleaved ATF6α acts as a transcription factor that enhances the expression of ER chaperones, autophagy-related genes, and ERAD-related genes. IRE1 activation induces the splicing of X-box binding protein 1 (XBP1) mRNA, and the spliced XBP1 acts as a transcription factor that increases the expression of genes that enhance protein folding and protein degradation. The UPR coordinates a transcription program that is the principal mechanism for maintaining ER homeostasis and a major mechanism for maintaining cellular proteostasis [1,2,3,4].

Many studies show that the UPR is a major player in regulating neuron viability and function in various neurodegenerative diseases. Nevertheless, these studies are highly contradictory. Some studies show that the UPR preserves neuron viability and function in neurodegenerative diseases, but other studies show opposite results [5,6,7,8,9,10]. While data indicate activation of the UPR in neurons under normal conditions [11,12], deficiency of individual branches of the UPR (either PERK, ATF6α, or IRE1) has a minimal effect on brain neurons in animals [13,14,15,16]. The minimal effect of deficiency of individual branches of the UPR on neurons is likely due to their functional redundancy [1,2,4,17]. It remains unclear whether or how the UPR participates in regulating neuronal proteostasis under normal and disease conditions.

It is believed that neuronal accumulation of hyperphosphorylated tau (p-tau) contributes significantly to neuron dysfunction and death in Alzheimer’s disease (AD) and other tauopathies, including progressive supranuclear palsy (PSP), frontotemporal dementias, among others [18,19,20]. Interestingly, genome-wide association studies (GWAS) show that polymorphism of the *Perk* gene is associated with an increased risk for PSP and an increased risk for late-onset AD in patients carrying APOE ε4 [21,22,23]. A recent study demonstrates that tauopathy-associated *Perk* alleles are functional hypomorphs with impaired PERK activity [24]. Moreover, an autopsy case report shows that a 4-year-old child with Wolcott-Rallison syndrome (caused by mutations in the *Perk* gene) exhibits evidence of the autophagy-lysosome pathway (ALP) impairment and neuronal p-tau accumulation in the central nervous system (CNS) [25]. Data indicate that PERK activation modulates the ALP activity in cells [26,27,28,29]. The ALP is one of the major mechanisms responsible for tau degradation in neurons [19,30,31]. Evidence suggests that impairment of the ALP induces neuronal accumulation of p-tau under normal and disease conditions [31,32,33,34]. These data imply a potential linkage among the UPR, ALP, and tau homeostasis in neurons under physiological conditions.

Our recent study shows that impairment of the UPR in oligodendrocytes via double deletion of PERK and ATF6α has no effect on actively myelinating oligodendrocytes in young developing mice but leads to the ALP impairment, intracellular accumulation of proteolipid protein, and apoptosis of mature oligodendrocytes in young adult mice [17]. Thus, to determine the physiological role of the UPR in neurons, we generated mice with double deletion of PERK and ATF6α in neurons. We found that the inactivation of PERK and ATF6α in neurons caused lysosomal dysfunction, impairment of autophagic flux, accumulation of p-tau and Aβ42, and hippocampal degeneration in adult mice. This finding defines the essential role of the UPR in maintaining the ALP function, proteostasis, and the viability and function of hippocampal neurons under physiological conditions.

## 2. Results

### 2.1. Inactivation of PERK and ATF6α in Neurons Led to Death and Memory Loss in Adult Mice

We have generated *PERK^loxP/loxP^; Thy1/CreER^T^*^2^ mice and demonstrated that tamoxifen treatment leads to PERK deletion selectively and ubiquitously in neurons in the CNS of *PERK^loxP/loxP^; Thy1/CreER^T^*^2^ mice [16]. Under physiological conditions, the basal activity of PERK in the CNS is barely detectable. PERK deletion, specifically in neurons, does not alter the levels of phosphorylated eIF2α and ATF4 but moderately reduces the level of CHOP. In contrast, the PERK-eIF2α pathway is activated in neurons of mice undergoing experimental autoimmune encephalomyelitis (EAE, an animal model of multiple sclerosis). PERK deletion, specifically in neurons, reduces phosphorylated eIF2α, ATF4, and CHOP levels in neurons during EAE. We have also shown that PERK inactivation in neurons does not affect their viability or function under normal conditions but exacerbates neurodegeneration in the animal model of multiple sclerosis [16]. Moreover, we have obtained *ATF6α* knockout (*ATF6α−/−*)mice that possess deletion of exons 8 and 9 of the *ATF6α* gene [35]. Under physiological conditions, the basal activity of ATF6α in the CNS is barely detectable. ATF6α is undetectable in the CNS of *ATF6α −/−* mice; however, global ATF6α deletion does not alter the level of BiP in the CNS (the target gene of ATF6α) [36]. In contrast, ATF6α is activated in the CNS of mice undergoing EAE, and global ATF6α deletion reduces the level of BiP in the CNS during EAE [36]. Data from our lab and other groups have demonstrated that global ATF6α deletion has no detectable effect on the brain under normal conditions but exacerbates brain pathology under disease conditions [14,36,37]. To determine the physiological role of the UPR in neurons, we generated mice with double deletion of PERK and ATF6α in neurons. *PERK^loxP^* mice were crossed with *Thy1/CreER^T^*^2^ mice, and the resulting progeny were crossed with *ATF6α−/−* mice to obtain *ATF6α +/−; PERK^loxP^*; *Thy1/CreER^T^*^2^ mice and *ATF6α+/−; PERK^loxP^* mice. *ATF6α+/−; PERK^loxP^; Thy1/CreER^T^*^2^ mice were further crossed with *ATF6α +/−; PERK^loxP^* mice to obtain *ATF6α−/−; PERK^loxP/loxP^; Thy1/CreER^T2^* mice and *PERK^loxP/loxP^; Thy1/CreER^T^*^2^ mice. One group of 8-week-old *ATF6α−/−; PERK^loxP/loxP^; Thy1/CreER^T^*^2^ mice were given i.p. injections of tamoxifen daily for eight consecutive days (Double KO mice). Another group of *ATF6α−/−; PERK^loxP/loxP^; Thy1/CreER^T^*^2^ mice were treated with vehicle (ATF6 KO mice). One group of 8-week-old *PERK^loxP/loxP^; Thy1/CreER^T^*^2^ mice were given i.p. injecting tamoxifen daily for eight consecutive days (PERK KO mice). Another group of *PERK^loxP/loxP^; Thy1/CreER^T^*^2^ mice, were treated with vehicle (WT mice). Interestingly, RT-PCR analysis showed that spliced XBP1 (XBP1s) mRNA was dramatically increased in the brain of Double KO mice compared to WT mice, PERK KO mice, and ATF6 KO mice (Figure 1A). These data suggest that inactivation of both the PERK and ATF6α branches of the UPR in neurons leads to impairment of the UPR and subsequently results in disruption of ER homeostasis and compensatory activation of the IRE1-XBP1 branch.

As expected, young adult PERK KO mice and ATF6 KO mice appeared healthy and were indistinguishable from WT mice. Although the body weight of Double KO mice was comparable with WT mice, PERK KO mice, and ATF6 KO mice (Figure 1B), greater than 50% of Double KO mice died starting as early as post-injection (of tamoxifen) day (PID) 6 (Figure 1C). While a few dead Double KO mice displayed epileptic seizures shortly before death, most died Double KO mice did not exhibit any noticeable neurological phenotypes. The sudden death of Double KO mice prohibited us from obtaining the CNS tissues to investigate the cause of death. Therefore, we focused on studying the surviving Double KO mice after that.

Hindlimb clasping reflex is a neurological sign of disease progression in several mouse models of neurodegeneration, including AD, Parkinson’s disease (PD), and cerebellar ataxia [38,39,40,41,42]. As expected, none of the PERK KO mice, ATF6 KO mice, and WT mice developed hindlimb clasping reflexes. Interestingly, all the surviving Double KO mice developed hindlimb clasping reflexes by PID30 (Figure 1D, E). Moreover, the Barnes maze test was performed to evaluate spatial learning and memory in these mice. Mice underwent four daily trials starting at PID 40 for four consecutive days. The memory retention (probe test) was performed 24 h following the fourth day of acquisition. During the trials, Double KO mice took a similar amount of time to find the target hole compared to WT mice, PERK KO mice, and ATF6 KO mice (Figure 2A). Notably, during the probe test, Double KO mice had significantly longer latency time to reach the closed target hole (Figure 2B), spent significantly less time in the Goal Zone of the maze (Figure 2C), and stayed at a longer average distance to the escape hole area (Figure 2D) than WT mice, PERK KO mice, and ATF6 KO mice. Conversely, the mean speed (Figure 2E) of Double KO mice was not reduced compared to WT mice, PERK KO mice, and ATF6 KO mice. These results suggest that the inactivation of PERK and ATF6α in neurons does not affect motor activity but causes spatial memory impairment in adult mice.

Hippocampal long-term potentiation (LTP) is the major mechanism for learning and memory [43]. To evaluate hippocampal LTP, electrophysiological studies were performed to examine hippocampal synaptic activities using acute brain slices from WT mice, PERK KO mice, ATF6 KO mice and Double KO mice at PID20 as described in our previous papers [44,45]. While hippocampal LTP of PERK KO mice and ATF6 KO mice were similar to WT mice, hippocampal LTP is impaired in Double KO mice (Figure 3A–C). These data suggest that the inactivation of PERK and ATF6α in neurons impairs hippocampal LTP in adult mice.

### 2.2. Inactivation of PERK and ATF6α in Neurons Led to Brain Atrophy and Hippocampal Degeneration in Adult Mice

We performed histopathological analysis to determine whether inactivation of PERK and ATF6α in neurons causes neurodegeneration in adult mice. Brian tissues were prepared from WT mice, PERK KO mice, ATF6 KO mice, and Double KO mice at PID 60. Whole brain scanning images of Nissl staining revealed severe atrophy of the hippocampus and modest atrophy of other brain regions (such as the cerebral cortex) in Double KO mice compared to WT mice, PERK KO mice, and ATF6 KO mice (Figure 4A). Accordingly, the brain weight of Double KO mice was significantly reduced at PID60 compared to WT mice, PERK KO mice, and ATF6 KO mice (Figure 4B).

High magnification images of Nissl staining showed severe hippocampal neuron loss in Double KO mice at PID60 compared to WT mice, PERK KO mice, and ATF6 KO mice (Figure 5(A1–A4)). Quantitative NeuN IHC showed dramatic neuron loss in the CA1 layer, severe neuron loss in the CA2 layer and CA3 layer, and significant neuron loss in the DG of Double KO mice at PID60 compared to WT mice, PERK KO mice and ATF6 KO mice (Figure 5(B1–B4),F). Moreover, quantitative NeuN IHC showed significant neuron loss in the layer V of the primary motor cortex in Double KO mice at PID60 compared to WT mice, PERK KO mice, and ATF6 KO mice (Figure 5(C1–C4),G). Additionally, CD11b (a marker for microglia) IHC showed noticeable activation of microglia in the hippocampus and cerebral cortex of Double KO mice at PID60 compared to WT mice, PERK KO mice, and ATF6 KO mice (Figure 5(D1–D4)). GFAP (a marker for astrocytes) also showed noticeable activation of astrocytes in the hippocampus and cerebral cortex of Double KO mice at PID60 compared to WT mice, PERK KO mice, and ATF6 KO mice (Figure 5(E1–E4)). Together, these results demonstrate that the inactivation of PERK and ATF6α in neurons led to brain atrophy and hippocampal degeneration in adult mice.

### 2.3. Inactivation of PERK and ATF6α in Neurons Led to Accumulation of p-tau and Aβ42 in the Hippocampus of Adult Mice

As described above, evidence suggests a potential linkage among impaired UPR, p-tau accumulation, and neurodegeneration. Thus, we determined if the inactivation of PERK and ATF6α induces p-tau accumulation in neurons. Interestingly, IHC using the AT8 antibody (that recognizes phosphorylated Ser202 and Thr205 in tau) revealed marked accumulation of p-tau in neurons in the hippocampus and cerebral cortex of Double KO mice at PID 60, as compared to WT mice, PERK KO mice, and ATF6 KO mice (Figure 6A1–A4,F). IHC using the CP13 antibody (that recognizes phosphorylated Ser202 in tau) (Figure 6B1–B4,G) and PHF1 (that identifies phosphorylated Ser396 and 404 in tau) (Figure 6C1–C4,H) also revealed noticeable accumulation of p-tau in neurons in the hippocampus and cerebral cortex of Double KO mice at PID 60, as compared to WT mice, PERK KO mice, and ATF6 KO mice. Accordingly, western blot using the AT8 antibody showed that the level of p-tau was markedly increased in the forebrain of Double KO mice at PID 60, as compared to WT mice, PERK KO mice, and ATF6 KO mice (Figure 7A,B). Conversely, western blot using the AT5 antibody (that recognizes total tau) showed that the level of total tau was comparable in the forebrain of WT mice, PERK KO mice, ATF6 KO mice, and Double KO mice at PID 60 (Figure 7A,C). Moreover, Aβ IHC using the 4G8 antibody showed that the level of Aβ was markedly elevated in the hippocampus and cerebral cortex of Double KO mice at PID 60, as compared to WT mice, PERK KO mice, and ATF6 KO mice (Figure 6D1–D4,I). Additionally, Aβ42 IHC using the Mab13.1.1 antibody showed that the level of Aβ42 was markedly elevated in the hippocampus and cerebral cortex of Double KO mice at PID 60, as compared to WT mice, PERK KO mice, and ATF6 KO mice (Figure 6E1–E4,J). These results suggest that the inactivation of PERK and ATF6α in neurons leads to the accumulation of p-tau and Aβ42 in adult mice’s hippocampus and cerebral cortex.

### 2.4. Inactivation of PERK and ATF6α Suppressed Lysosomal Function and Impaired Autophagic Flux in Hippocampal Neurons of Adult Mice

Data indicate that the UPR modulates the activity of the ALP by regulating the transcription of genes related to the ALP [29,46,47]. Interestingly, our recent study showed that inactivation of PERK and ATF6α in oligodendrocytes attenuates the transcription of SEZ612 and GNPTAB (which participate in the transport of cathepsin D from the trans-Golgi network to the lysosome), leads to delocalization of cathepsin D from the lysosome, and results in impairment of autophagic flux in mature oligodendrocytes of young adult mice [17]. We thus determined the role of the inactivation of PERK and ATF6α in the ALP in neurons. It is known that impairment of autophagic flux induces accumulation of ubiquitinated proteins and p62 (sequestosome-1) in neurons [31,33,48,49]. p62 (an autophagic cargo adaptor) can bind to ubiquitinated and autophagosome membrane proteins and deliver ubiquitinated proteins to the lysosome for degradation through the ALP. p62 itself is also degraded in the autolysosome. Accumulation of p62 indicates impairment of autophagic flux [50,51,52]. Interestingly, IHC using a pan ubiquitin antibody showed that the immunoreactivity of ubiquitin was markedly increased in neurons in the hippocampus and cerebral cortex of Double KO mice at PID 60, as compared to WT mice, PERK KO mice, and ATF6 KO mice (Figure 8A1–A4,C). p62 IHC showed that the immunoreactivity of p62 was markedly increased in neurons in the hippocampus and cerebral cortex of Double KO mice at PID 60, as compared to WT mice, PERK KO mice, and ATF6 KO mice (Figure 8B1–B4,D). Currently, LC3 is the most widely used autophagosome marker [53,54]. Nascent LC3 is processed at its C terminus by ATG4 and becomes LC3-I, and then LC3-I is conjugated with phosphatidylethanolamine to become LC3-II, which associates with the autophagosome membranes. After fusion with the lysosome, LC3-II is degraded in the autolysosome. The amount of LC3-II is believed to reflect the number of autophagosomes and autophagy-related structures [52]. Importantly, western blot analysis showed the ratio of LC3-II/LC3-I was significantly increased in the forebrain of Double KO mice at PID 60, as compared to WT mice, PERK KO mice, and ATF6 KO mice (Figure 8E,F). The elevated levels of LC3-II, p62, and ubiquitinated proteins in neurons of Double KO mice suggest that the inactivation of PERK and ATF6α impairs autophagic fons in the hippocampus and cerebral cortex.

Cathepsin D, the major lysosomal aspartyl protease, is synthesized in the ER, and transported to the lysosome via the trans-Golgi network [55,56]. Under acidic conditions in the lysosome, a mature single-chain form of cathepsin D (46 kDa) is processed into active two-chain forms of cathepsin D (31 and 14 kDa). Interestingly, western blot showed that the level of the mature single-chain form of cathepsin D was significantly increased in the forebrain of Double KO mice at PID 60, as compared to WT mice, PERK KO mice, and ATF6 KO mice (Figure 8E,G). It is known that v-ATPase is a proton pump that acidifies the newly generated lysosome [57,58]. Importantly, western blot showed that the protein level of the V0a1 subunit of v-ATPase was dramatically decreased in the forebrain of Double KO mice at PID 60, as compared to WT mice, PERK KO mice, and ATF6 KO mice (Figure 8E,H). Acidification of the lysosome is necessary for activating cathepsins and degradation of proteins delivered to the lysosome via the ALP [55,56,57,58]. Thus, these data suggest the possibility that the inactivation of PERK and ATF6α causes lysosomal acidification defect by suppressing the expression of v-ATPase V0a1, leading to impaired degradative function of the lysosome and accumulation of cathepsin D in the lysosome, and results in impairment of autophagic flux in neurons in the hippocampus and cerebral cortex.

## 3. Discussion

A hallmark characteristic of neurodegenerative diseases is aggregation of misfolded proteins in neurons resulting from disruption of cellular proteostasis [59,60,61]. Cells possess a collection of highly conserved mechanisms (referred to as the cellular proteostasis network), including the cytosolic heat-shock response (HSR), the UPR, ERAD, the ubiquitin-proteasome system (UPS), and the ALP, which are integral in safeguarding cellular proteome integrity and maintaining cellular proteostasis [59,61]. Many studies show that neurons are highly sensitive to disruption of ER protein homeostasis and that the UPR is a major player in regulating neuron viability and function in various neurodegenerative diseases, including AD and PD [5,6,7,9]. Recent studies suggest that the UPR has essential physiological functions in brain development, neuronal differentiation, synaptic plasticity, and memory storage [12]. Nevertheless, deficiency of individual branches of the UPR (either PERK, ATF6α, or IRE1) has a minimal effect on brain neurons in animals [13,14,15,16]. It remains unknown whether or how the UPR participates in regulating neuronal proteostasis under normal and disease conditions. We believe that the minimal effect of deficiency of individual branches of the UPR on neurons is due to their functional redundancy. Thus, to determine the physiological role of the UPR in neurons, we generated mice with global ATF6α deletion and neuron-specific PERK deletion. Importantly, we found that inactivation of PERK and ATF6α in neurons caused lysosome dysfunction (as evidenced by decreased expression of the V0a1 subunit of v-ATPase and accumulation of cathepsin D), impairment of autophagic flux (as evidenced by increased ratio of LC3-II/LC3-I and increased p62 level), accumulation of p-tau and Aβ42, and hippocampal degeneration in adult mice. These findings suggest that the UPR is required to maintain proteostasis in hippocampal neurons and preserve their viability and function under physiological conditions.

Several studies showed that the UPR is critical in regulating neuron viability and function in AD and other tauopathies [5,6,7,9]. Importantly, recent studies show that polymorphism in the *Perk* gene, associated with reduced PERK function, increases susceptibility to PSP and AD [21,22,23,24]. We showed here that inactivation of PERK and ATF6α in neurons led to impairment of spatial memory, impairment of hippocampal LTP, accumulation of Aβ42 and p-tau in the hippocampus, and neurodegeneration in the hippocampus of adult mice, which recapitulate multiple features of AD in mice. Moreover, it is known that both the UPR activity and hippocampal function decline with age [59,62,63,64], and that age is the single biggest risk factor for AD and other neurodegenerative diseases [65,66]. Collectively, these data suggest the possibility that impairment of the UPR with age contributes to disruption of neuronal proteostasis, accumulation of Aβ42 and p-tau, and dysfunction and death of hippocampal neurons in normal aging as well as AD and other neurodegenerative diseases.

We showed here that adult mice with inactivation of PERK and ATF6α in neurons displayed lysosomal dysfunction, impairment of autophagic flux, p-tau accumulation, and neurodegeneration in the hippocampus (including the CA1 layer, CA2 layer, CA3 layer, and DG) and cerebral cortex. It is well documented that lysosomal dysfunction in neurons impairs autophagic flux, leads to intercellular accumulation of lipids and/or proteins, and subsequently results in neurodegeneration [67,68]. A recent report shows that mice with deletion of v-ATPase V0a1 in neurons exhibit selective degeneration of the CA1 layer in the hippocampus (the CA3 layer and DG remain largely intact). However, this report falls short of examining the impact of deletion of v-ATPase V0a1 on lysosomal function and autophagic flux in neurons [69]. The ALP is one of the major mechanisms responsible for tau degradation in neurons [19,30,31]. Impairment of the ALP induces neuronal accumulation of p-tau under normal and disease conditions [31,32,33,34]. Moreover, neuronal accumulation of p-tau can lead to hippocampal degeneration [18,19,20]. Thus, these data suggest the possibility that lysosomal dysfunction resulting from the decreased expression of v-ATPase V0a1 contributes to p-tau accumulation and hippocampal degeneration in adult mice with PERK and ATF6α deletion in neurons. Conversely, the much more profound neuropathology displayed by Double KO mice vs. neuron-specific v-ATPase V0a1 KO mice indicates that other mechanisms, besides the decreased expression of v-ATPase V0a1, also contribute to the detrimental effects of inactivation of PERK and ATF6α on neurons in adult mice.

In summary, we generated mice with double deletion of PERK and ATF6α in neurons. We showed that these mice displayed impairment of spatial memory, impairment of hippocampal LTP, disruption of the ALP, accumulation of p-tau and Aβ42, and hippocampal degeneration in adult mice. This finding defines the essential role of the UPR in maintaining the ALP function, proteostasis (especially tau and Aβ homeostasis), and the viability and function of hippocampal neurons under physiological conditions. On the other hand, it is known that both the UPR activity and hippocampal function decline with age and that age is the single biggest risk factor for AD and other neurodegenerative diseases. As such, this finding also suggests the involvement of UPR impairment in hippocampal impairment in normal aging and AD and other neurodegenerative diseases.

## 4. Materials and Methods

### 4.1. Mice and Tamoxifen Treatment

We obtained *ATF6α* knockout (*ATF6α−/−*) mice [35] from Dr. Kazutoshi Mori (Kyoto University, Kyoto, Japan). We purchased *PERK^loxP^* mice (stock number 023066) that possess loxP sites flanking exons 3–5 of the *Perk* gene [70] and *Thy1/CreER^T^*^2^ transgenic mice (stock number 023066) that express the CreER^T*2*^ recombinase under the control of the Thy1.2 promoter [71] from the Jackson Laboratory. *PERK^loxP^* mice were crossed with *Thy1/CreER^T^*^2^ mice, and the resulting progeny were crossed with *ATF6α−/−* mice to obtain *ATF6α+/−; PERK^loxP^*; *Thy1/CreER^T^*^2^ mice and *ATF6α+/−*; *PERK^loxP^* mice. *ATF6α+/−; PERK^loxP^; Thy1/CreER^T^*^2^ mice were further crossed with *ATF6α +/−; PERK^loxP^* mice to obtain *ATF6α−/−; PERK^loxP/loxP^; Thy1/CreER^T^*^2^ and *PERK^loxP/loxP^; Thy1/CreER^T^*^2^ mice. To induce CreER^T2^-mediated recombination, 8-week-old *ATF6α−/−; PERK^loxP/loxP^; Thy1/CreER^T^*^2^ mice and *PERK^loxP/loxP^; Thy1/CreER^T^*^2^ mice were given i.p. injections of tamoxifen (1 mg/day in corn oil) or vehicle daily for eight consecutive days. Genotypes were determined by PCR from DNA extracted from tail tips as described in previous papers [35,70,71]. Both male and female mice were used in this study, and no sex-based differences were observed. All animal procedures were conducted in complete compliance with the NIH Guide for Care and Use of Laboratory Animals and were approved by the Institutional Animal Care and Use Committee of the University of Minnesota.

### 4.2. Hindlimb Clasping Test

Mice were suspended by the base of the tail and observed for 10–15 s. Three separate trials were taken each day for each mouse. One or two hindlimbs retracted inwards towards the abdomen for at least 50% of the observation period is recorded as hindlimb clasping test positive.

### 4.3. Immunohistochemistry (IHC) and Nissl Staining

Anesthetized mice were perfused through the left cardiac ventricle with 4% paraformaldehyde in PBS. Half sagittal were post-fixed in 4% paraformaldehyde for 2 h, cryoprotected in 30% sucrose for 48 h, embedded in optimum cutting temperature compound and frozen on dry ice. Frozen sections were cut using a cryostat at a thickness of 10 µm. The other half sagittal brain was post-fixed in 4% paraformaldehyde for 72 h, dehydrated through graded alcohols and embedded in paraffin wax. Paraffin sections were cut using a microtome at a thickness of 5 µm. Immunohistochemical detection of NeuN (1:100, Millipore, RRID:AB_2313673), GFAP (1:100; COVΛNCE, LN#14831701), CD11b (1:50; Millipore, RRID:AB_92930), ubiquitin (P4D1, 1:1000; Santa Cruz, sc8017), p62 (1:15,000; Abcam, catalog ab56416), CP13 (1:200; recognizes phosphorylated Ser202 of the tau protein, provided by Dr. Peter Davis at Alber Einstein College of Medicine, New York, NY, USA), PHF1 (1:1000, recognizes phosphorylated Ser396 and 404 of the tau protein, provided by Dr. Peter Davis at Alber Einstein College of Medicine, New York, NY, USA), AT8 (1:50; recognizes phosphorylated Ser202 and Thr205 of the tau protein, MN1020, ThermoFisher Scientific, Waltham, MA, USA), Mab13.1.1 (1:1000; recognizes Aβ42, provided by Dr. Pritam Das at Mayo Clinic, Jacksonville, FL, USA), and 4G8 (1:2000; recognizes Aβ, Covance, San Carlos, CA, USA) were performed as described in previous papers [16,72,73]. Immunopositive cells were counted as described in our previous paper [16].

Paraffin sections were used for Nissl staining. After deparaffinization, the sections were stained with 0.1% cresyl viole solution for 3 min, rinsed in water, differentiated in 95% alcohol for 3 min, dehydrated in 100% alcohol, and cleared in xylene. Stained brain sections were scanned by a TissueScope LE120 slide scanner (Huron Digital Pathology, St. Jacobs, ON, Canada).

For quantification of neurons in the hippocampus, 5 µm thick sagittal brain sections were cut, and every tenth sagittal section containing the dorsal hippocampus in the series spanning from Bregma lateral 1.08 mm to 1.56 mm was immunoassay with the NeuN antibody. To quantify neurons in layer V of the primary motor cortex, 5 μm thick serial sections were cut from Bregma lateral 1.08 mm to 1.32 mm. Every tenth section was stained with NeuN antibody. The total NeuN-positive neurons in the CA1 layer, CA2 layer, CA3 layer, dentate gyrus (DG), and layer V of the primary motor cortex were counted and analyzed using the NIH ImageJ software(https://imagej.nih.gov).

### 4.4. Barnes Maze Learning and Memory Test

Spatial learning and memory were evaluated using the Barnes maze test described in our previous papers [44,45]. A Barnes maze with a video-tracking system was purchased from San Diego Instrument. ANYmaze video-tracking software (https://www.any-maze.com, Stoelting™) was used for behavioral analysis. The maze consists of 20 exploration holes, with only one hole leading to a recessed escape box during task acquisition on an elevated platform. In each trial, one mouse was first placed under a box in the center of the maze for 15 s and then allowed to freely explore the maze to search for the escape hole (target) for 3 min after the removal of the box. An escape from the maze was defined as the movement of the mouse completely through the escape hole into the recessed box. In the acquisition period (learning phase), the mouse underwent four daily trials with an intertrial interval of 25–30 min for four consecutive days. The memory retention (probe test) was performed 24 h after the fourth day of acquisition by covering all holes and occupancy plots as the exploration pattern for each group of mice was determined. Retention of memory was measured by quantifying the time the mouse spent in the target zone, the first time the mouse reached the escape hole area, and the distance from the animal to the position of the removed escape hole (the target) during this 90 s probe test.

### 4.5. XBP1 Splicing Assay

RNA was isolated from the brain using TRIzol reagent (Thermo Fisher Scientific, Waltham, MA, USA) following the manufacturer’s instructions and treated with DNase I (Thermo Fisher Scientific, Waltham, MA, USA) to eliminate genomic DNA. Reverse transcription was performed using the iScript cDNA Synthesis Kit (Bio-Rad Laboratories, Hercules, CA, USA). PCR for XBP1 was performed to detect XBP1s mRNA using Taq DNA Polymerase (QIAGEN). PCR products were separated by electrophoresis on a 3% agarose gel, as described in our previous papers [17,36].

### 4.6. Western Blot Analysis

Half forebrain was harvested from mice, rinsed in ice-cold PBS, and homogenized in RIPA buffer (50 mM Tris-HCl pH 7.4, 150 mM NaCl, 0.5% Triton X, 1 mM EDTA, 0.3% SDS, 1% sodium deoxycholate) with protease and phosphatase inhibitors (0.1 mM phenylmethylsulfonyl fluoride, protease inhibitor cocktail (P8340, Sigma-Aldrich, St. Louis, MO, USA); phosphatase inhibitor cocktails (P2850 and P5726, Sigma-Aldrich, St. Louis, MO, USA) and 0.2 mM 1,10-phenanthroline monohydrate). Soluble and insoluble proteins were extracted by drawing up and expulsing tissue through 1 mL MonojectTM syringes (Covidien, Costa Mesa, CA) first without and then with 20 G BD PrecisionGlideTM needles. Homogenates were incubated at 4 °C for 1 h, then centrifuged at 13,000 rpm for 90 min at 4 °C, and the supernatant was collected. The protein concentration was determined using the DC Protein Assay (Bio-Rad Laboratories, Hercules, CA, USA). The extracts (120 µg) were separated on SDS-PAGE gels and transferred to nitrocellulose membranes. For Tau analysis, the membranes were incubated with primary antibodies against TAU5 (a pan tau antibody, 1:30,000, MAB361, Millipore, Burlington, MA, USA), AT8 (1:1000, MN1020, Thermo Fisher, Waltham, MA, USA), or GAPDH (14C10, 1:4000, Cell Signaling Technology, Danvers, Massachusetts, USA) overnight, followed by IRDye-linked goat anti-mouse 680LT (red) and goat anti-rabbit 800CW (green) secondary antibodies (1:100,000 dilution LI-COR Biosciences, Lincoln, Nebraska, USA). Immunoreactivity was visualized using a LiCor imaging system and Image Studio software (https://www.licor.com/bio/image-studio, Odyssey). For Autophagy analysis, 10 µg protein was separated by SDS-PAGE followed by Bio-Rad Trans-Blot Turbo transfer using nitrocellulose membranes and then incubated with primary antibodies against LC3 (1:1000, 2775, Cell Signaling, Danvers, MA, USA), Cathepsin D (1:1000, Abcam, ab75852,) or the V0a1 subunit of vacuolar [H+] ATPase (v-ATPase) (1:500, Abcam, ab105937). An HRP-conjugated secondary antibody (1:1000, anti-mouse, catalog PI-2000; anti-rabbit, PI-1000, Vector Laboratories, Newark, CA, USA) was used in this study. The chemiluminescent signal was detected using the ECL Detection Reagents (GE Healthcare Biosciences, Piscataway, NJ, USA). The intensity of the chemiluminescence signals was quantified using the NIH ImageJ software (https://imagej.nih.gov/).

### 4.7. Long-Term Potentiation (LTP) Experiment

Hippocampal slice preparation: Coronal hippocampal slices were obtained from mice post-injection of tamoxifen day (PID) 20. Animals were decapitated. The brain was rapidly removed and placed in ice-cold artificial cerebrospinal fluid (ACSF). Coronal slices (350 μm thick) were cut from a block of tissue containing the hippocampus using a vibratome (Microm HM 650 V, Thermo Scientific, Waltham, MA, USA). Slices were incubated (>30 min) at room temperature (21–24 °C) in ACSF containing: NaCl 124 mM, KCl 2.69 mM, KH_2_PO_4_ 1.25 mM, MgSO_4_ 2 mM, NaHCO_3_ 26 mM, CaCl_2_ 2 mM, and glucose 10 mM, and was gassed with 95% O_2_/5% CO_2_ (pH = 7.3–7.4). Slices were transferred to an immersion recording chamber and superfused at 2 mL/min with gassed ACSF, including 0.05 mM Picrotoxin and 1 µM (2S)-3-[[(1S)-1-(3,4-dichlorophenyl)ethyl]amino-2-hydroxypropyl](phenylmethyl)phosphonic acid hydrochloride (CGP54626) to block GABAA and GABAB receptors, respectively.

Electrophysiology: The field excitatory postsynaptic potentials (fEPSPs) were recorded with an EX-1 amplifier (Dagan Instruments, Minneapolis, MN, US), using pipettes (2–3 MU) filled with ACSF placed in the Stratum radiatum (s. r.) of the CA1 region. Signals were acquired with a 10 KHz sampling rate and filtered with a 3 Hz low pass filter and a 300 Hz high pass filter. Synaptic responses were evoked at 0.33 Hz by stimulation (2 ms duration) of the Schaffer Collaterals (CS) using a concentric bipolar electrode placed in the s. r. of the CA1 region. After a stable baseline of at least 15 min, LTP was induced by applying a tetanic stimulation (4 trains at 100 Hz for 1 s; 30 s intervals) in the SCs. fEPSP slope was normalized to 15 min of baseline recording. The pCLAMP 10.4 (Molecular Devices, San Jose, CA, USA) software was used for stimulus generation, data display, acquisition, and storage.

### 4.8. Statistics

Analyses were conducted using GraphPad Prism version 9 software. Data are presented as means ± s.e.m. Comparisons between 2 groups were statistically evaluated by a 2-tailed *t* test. Multiple comparisons were statistically evaluated by a one-way ANOVA with Tukey’s posttest. *p* values less than 0.05 were considered significant.

## Figures and Tables

**Figure 1 ijms-24-11542-f001:**
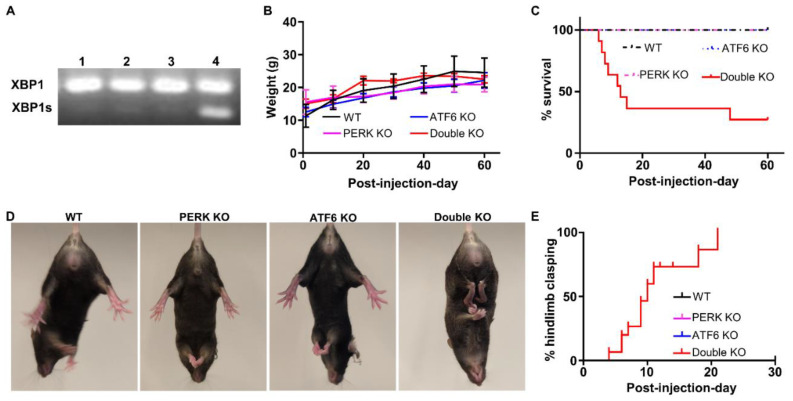
Inactivation of PERK and ATF6α in neurons led to death and abnormal hindlimb clasping reflex in mice. (**A**) PCR analysis showed that spliced XBP1 (XBP1s) mRNA was dramatically increased in the brain of Double KO mice (lane 4) compared to WT mice (lane 1), PERK KO mice (lane 2), and ATF6 KO mice (lane 3) at PID30. (**B**) Body weight of WT mice (*N* = 8 mice), PERK KO mice (*N* = 11 mice), ATF6 KO mice (*N* = 14 mice), and Double KO mice (*N* = 13 mice). (**C**) Survival curve of WT mice, PERK KO mice, ATF6 KO mice, and Double KO mice. *N* = 20 mice. (**D**) Double KO mice developed hindlimb clasping reflexes compared to WT mice, PERK KO mice, and ATF6 KO mice. (**E**) The percentage of WT mice (*N* = 9 mice), PERK KO mice (*N* = 9 mice), ATF6 KO mice (*N* = 9 mice), and Double KO mice (*N* = 15 mice) developed hindlimb clasping reflex.

**Figure 2 ijms-24-11542-f002:**
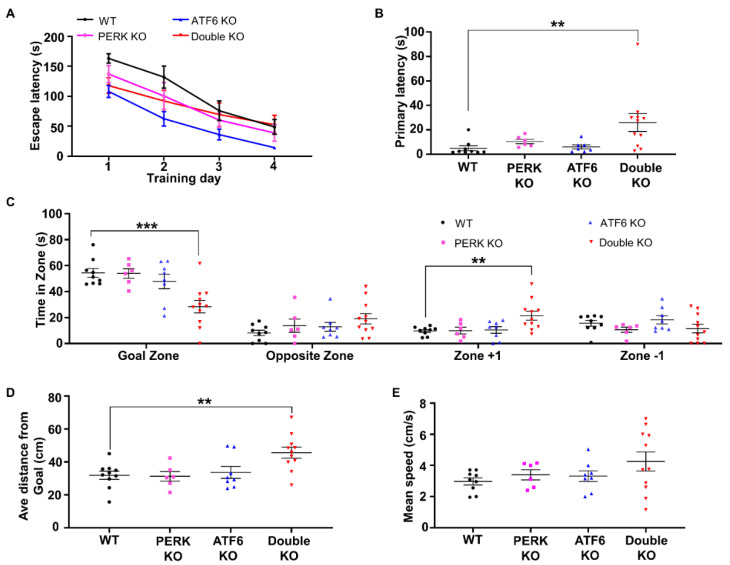
Inactivation of PERK and ATF6α in neurons led to memory loss. (**A**) Latency time to escape the maze during four consecutive training days. (**B**) The first time to reach the escape hole area was during the probe trial. (**C**) The time animals spent in each zone (Goal zone, Opposite zone, Zone+1 and Zone −1) of the maze during the probe trial. (**D**) The average distance from the animal to the position of the removed escape hole (the target) during the 90 s probe trial. (**E**) Mean speed during the probe trial. WT mice, *N* = 9 mice; PERK KO mice, *N* = 6 mice; ATF6 KO mice, *N* = 8 mice; Double KO mice, *N* = 11 mice. Data are presented as means ± s.e.m. ** *p* < 0.01, *** *p* < 0.001.

**Figure 3 ijms-24-11542-f003:**
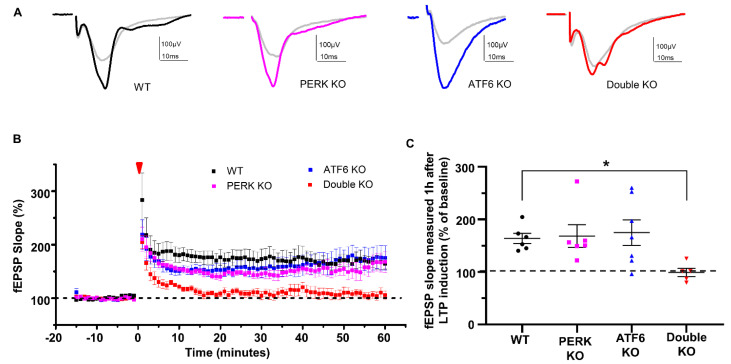
LTP was impaired in the hippocampus of Double KO mice. (**A**) Representative traces of the CA1 synaptic field potentials evoked by stimulation of the Shaffer collateral pathway in WT mice, ATF6 KO mice, PERK KO mice, and Double KO mice at PID20. Traces in gray color are average fEPSP prior to tetanic stimulation, others (black, magenta, blue and red color) are average fEPSP after tetanic stimulation. (**B**) fEPSP slope versus time in WT mice, ATF6 KO mice, PERK KO mice, and Double KO mice at PID20. 0 time and red arrowhead indicate tetanic stimulation. The baseline is set at 100%. (**C**) The percentage of potentiation of the fEPSP slope 1 h after stimulation in WT mice, ATF6 KO mice, PERK KO mice, and Double KO mice. WT mice, *N* = 6 mice; PERK KO mice, *N* = 6 mice; ATF6 KO mice, *N* = 7 mice; Double KO mice, *N* = 5 mice. Data are presented as means ± s.e.m. * *p* < 0.05.

**Figure 4 ijms-24-11542-f004:**
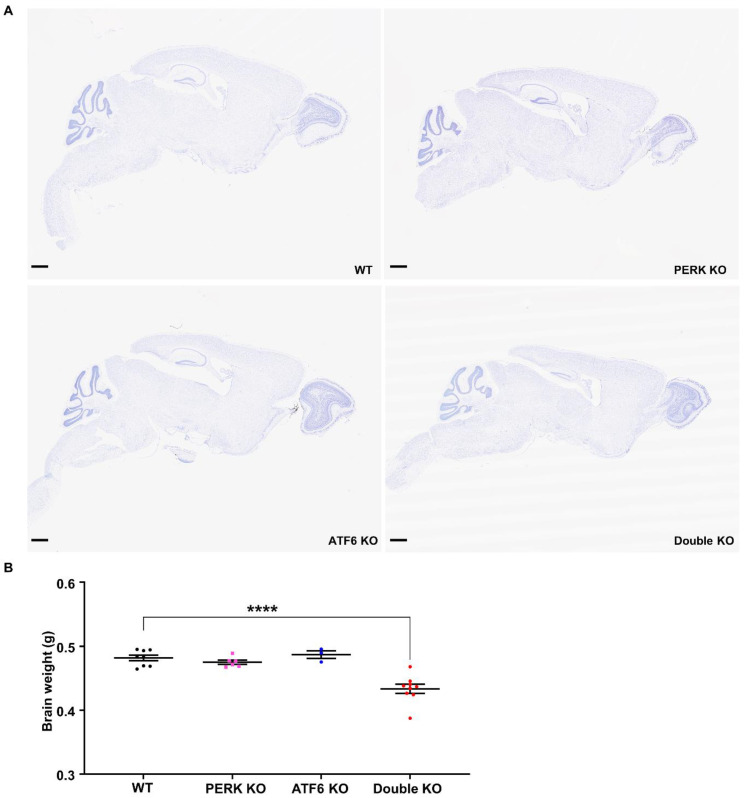
Inactivation of PERK and ATF6α in neurons led to brain atrophy and weight loss. (**A**) Representative images of Nissl staining of the whole brain of WT mice, ATF6 KO mice, PERK KO mice and Double KO mice at PID60. Scale bar, 1 mm. (**B**) Brain weight of WT mice (*N* = 8 mice), ATF6 KO mice (*N* = 3 mice), PERK KO mice (*N* = 6 mice), and Double KO mice (*N* = 9 mice) at PID60. Data are presented as means ± s.e.m. **** *p* < 0.0001.

**Figure 5 ijms-24-11542-f005:**
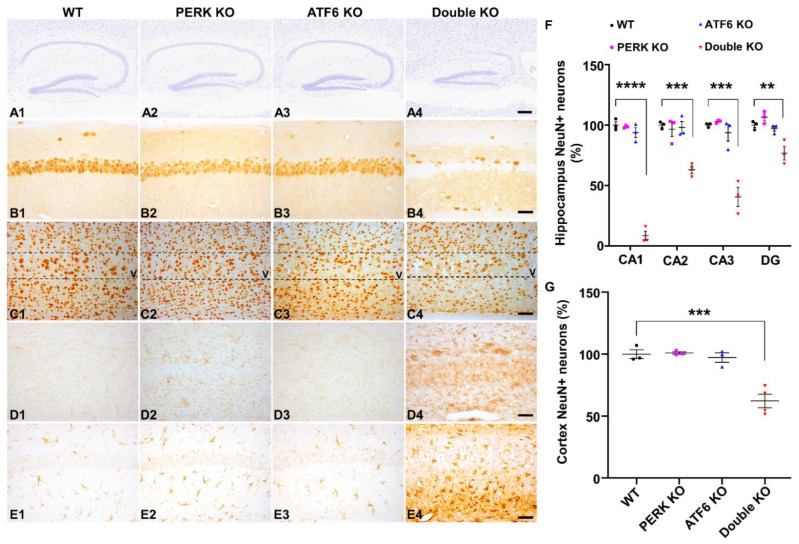
Inactivation of PERK and ATF6α in neurons led to neurodegeneration in the CNS. (**A1**–**A4**) Nissl staining of the hippocampus showed hippocampal atrophy in Double KO mice at PID60, compared to WT mice, ATF6 KO mice, and PERK KO mice. Scale bar, 200 μm. (**B1**–**B4**) NeuN IHC showed dramatic neuron loss in the hippocampal CA1 layer of Double KO mice at PID60 compared to WT mice, ATF6 KO mice, and PERK KO mice. Scale bar, 50 μm. (**C1**–**C4**) NeuN IHC showed moderate neuron loss in layer V of the primary motor cortex of Double KO mice at PID60 compared to WT mice, ATF6 KO mice, and PERK KO mice. Scale bar, 50 μm. (**D1**–**D4**) CD11B IHC showed microglia activation in the hippocampus of Double KO mice at PID60 compared to WT mice, ATF6 KO mice, and PERK KO mice. Scale bar, 50 μm. (**E1**–**E4**) GFAP IHC showed activation of astrocytes in the hippocampus of Double KO mice at PID60, as compared to WT mice, ATF6 KO mice, and PERK KO mice. Scale bar, 50 μm. (**F**) The relative NeuN positive neuron number in the CA1 layer, CA2 layer, CA3 layer, and DG of WT mice, ATF6 KO mice, PERK KO mice, and Double KO mice at PID60. *N* = 3 mice for each group. (**G**) The relative NeuN positive neuron number in the layer V of the primary motor cortex in WT mice (*N* = 3 mice), ATF6 KO mice (*N* = 3 mice), PERK KO mice (*N* = 3 mice), and Double KO mice (*N* = 4 mice) at PID60. Data are presented as means ± s.e.m. ** *p* < 0.01, *** *p* < 0.001, **** *p* < 0.0001.

**Figure 6 ijms-24-11542-f006:**
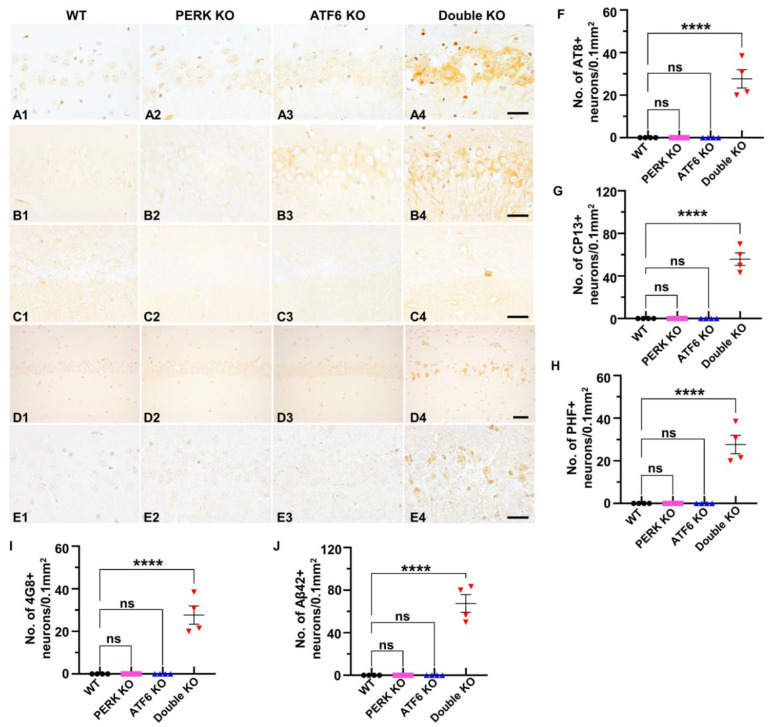
Inactivation of PERK and ATF6α in neurons led to the accumulation of p-tau and Aβ42 in the hippocampus. (**A1**–**A4**) AT8 IHC showed AT8-positive p-tau accumulation in the hippocampus of Double KO mice at PID60 compared to WT mice, ATF6 KO mice, and PERK KO mice. Scale bar, 20 μm. (**B1**–**B4**) CP13 IHC showed CP13-positive p-tau accumulation in the hippocampus of Double KO mice at PID60 compared to WT mice, ATF6 KO mice, and PERK KO mice. Scale bar, 20 μm. (**C1**–**C4**) PHF1 IHC showed PHF-positive p-tau accumulation in the hippocampus of Double KO mice at PID60 compared to WT mice, ATF6 KO mice, and PERK KO mice. Scale bar, 20 μm. (**D1**–**D4**) 4G8 IHC showed Aβ accumulation in the hippocampus of Double KO mice at PID60 compared to WT mice, ATF6 KO mice, and PERK KO mice. Scale bar, 50 μm. (**E1**–**E4**) Aβ42 IHC showed Aβ42 accumulation in the hippocampus of Double KO mice at PID60, compared to WT mice, ATF6 KO mice, and PERK KO mice. Scale bar, 20 μm. (**F**–**J**) Quantification of 4G8, CP13, PHF1, 4G8 and Aβ42 positive neurons in the hippocampus of WT mice, ATF6 KO mice, PERK KO mice, and Double KO mice. *N* = 4 mice for each group. Data are presented as means ± s.e.m. ns: no significance; **** *p* < 0.0001.

**Figure 7 ijms-24-11542-f007:**
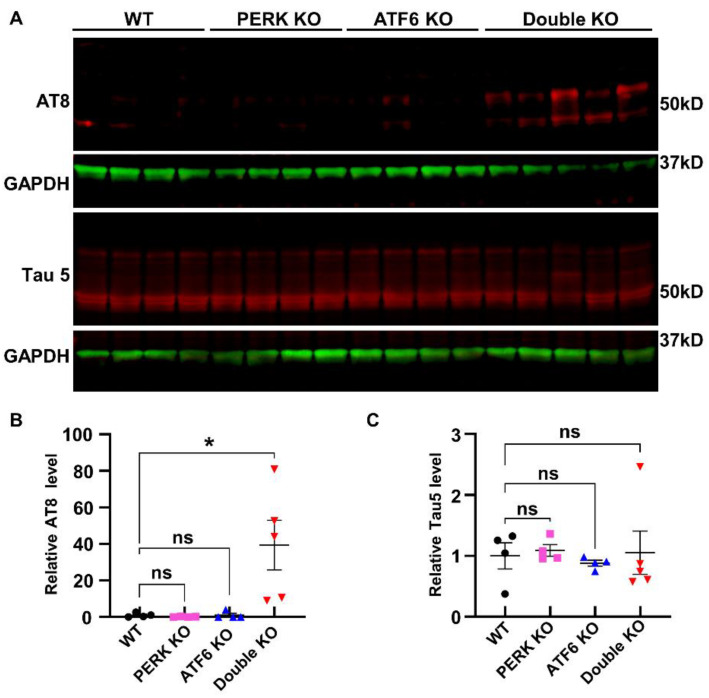
Inactivation of PERK and ATF6α in neurons led to the accumulation of p-tau in the forebrain of mice. Western blot showed that the elevated level of AT8-positive p-tau in the forebrain of Double KO mice compared to WT mice, ATF6 KO mice, and PERK KO mice (**A**,**B**), and a comparable level of total tau (**A**,**C**) in the forebrain WT mice (*N* = 4 mice), ATF6 KO mice (*N* = 4 mice), PERK KO mice (*N* = 4 mice), and Double KO mice (*N* = 5 mice) at PID60. Data are presented as means ± s.e.m. ns: no significance; * *p* < 0.05.

**Figure 8 ijms-24-11542-f008:**
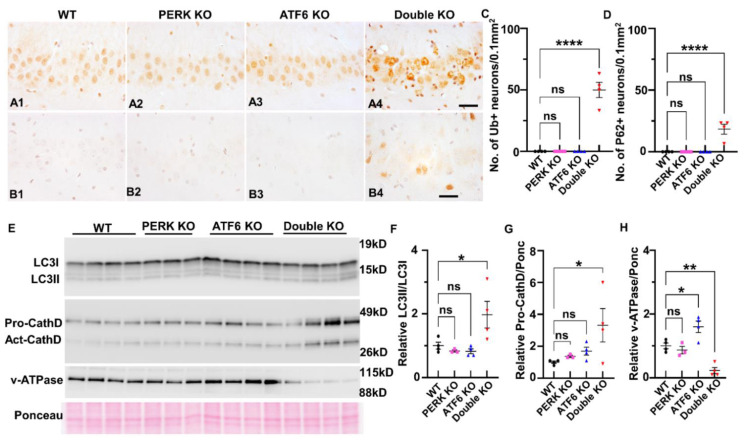
Inactivation of PERK and ATF6α led to impairment of the ALP. (**A1**–**A4**) ubiquitin IHC showed an elevated level of ubiquitin in the hippocampus of Double KO mice at PID60 compared to WT mice, ATF6 KO mice, and PERK KO mice. Scale bar, 20 μm. (**B1**–**B4**) p62 IHC showed an elevated level of p62 in the hippocampus of Double KO mice at PID60, as compared to WT mice, ATF6 KO mice, and PERK KO mice. Scale bar, 20 μm. (**C**,**D**) Quantification of Ubiquitin, and P62 positive neurons in the hippocampus of WT mice, ATF6 KO mice, PERK KO mice, and Double KO mice. N = 4 mice for each group. Data are presented as means ± s.e.m. ns: no significance; **** *p* < 0.0001. (**E**,**F**) Western blot showed the elevated level of LC3-II/LC3-I in the forebrain of Double KO mice at PID60, compared to WT mice, ATF6 KO mice, and PERK KO mice. (**E**,**G**) Western blot showed the elevated level of pro-cathepsin D (pro-CathD) in the forebrain of Double KO mice at PID60, compared to WT mice, ATF6 KO mice, and PERK KO mice. (**E**,**H**) Western blot showed the decreased level of ATPase V0a1 in the forebrain of Double KO mice at PID60, as compared to WT mice, ATF6 KO mice, and PERK KO mice. WT, *N* = 4 mice; PERK KO, *N* = 3 mice; ATF6 KO, *N* = 4 mice; Double KO, *N* = 4 mice. Data are presented as means ± s.e.m. ns: no significance; * *p* < 0.05, ** *p* < 0.01, **** *p* < 0.0001.

## Data Availability

All data generated or analyzed in this study are included in this paper and can be obtained from the authors upon reasonable request.

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
