# Peer review of "The UPR Maintains Proteostasis and the Viability and Function of Hippocampal Neurons in Adult Mice"

_ijms, 2023, doi:10.3390/ijms241411542_

Round 1

Reviewer 1 Report

This manuscript described the effects of disruption of unfolded protein response (UPR) by knocking out two of the three branches of UPR: PERK and ATF6α. The study employed WT, PERK KO, ATF6α KO, and Double KO mice of both sexes. The experiments are described in detail and logically presented. The results are presented mostly without ambiguity. Interpretations/conclusions drawn from the results are clear and consistent with the results. M & M section has detailed descriptions of animal breeding however, actual genotypes of animals used in the experiments are not listed. For example, what is the actual genotype(s) of animals shown as WT? The study employed two out of the three UPR branches but the rationale for this selection is not presented. IHC data in Fig. 6 are presented as photographs and no quantitative analyses are presented. Most conclusions seem to be correct, but some are not convincing. For example, this reviewer cannot see “marked accumulation of p-tau” by PHF1 IHC (Fig 6. C1-C4). Quantitative data and statistical analyses would make the conclusion stronger and more convincing. On pages 4 and 5, figure numbers are not shown in boldface. This manuscript should be accepted after minor revision.

Reviewer 2 Report

In this manuscript, Liu and colleagues investigated the role of the unfolded protein response (UPR) in neuronal homeostasis under physiological conditions. To this end, the authors generated double knockout (KO) mice harboring deletions for both PERK and ATF6α in neurons. Spliced XBP1 (XBP1s) mRNA was detected in these mice, suggesting that the IRE-XBP branch of the UPR was activated to compensate for the inactivation of PERK and ATF6α.

Double PERK and ATF6α KO mice were found to have longer latency time to reach the closest target hole compared to single PERK or ATF6α mutants. Similarly, hippocampal long-term potentiation (LTP) was also impaired in PERK/ATF-6 double KO mice.  By contrast, mean speed remained unaffected, which is indicative of normal motor activity.
Besides, inactivation of both PERK and ATF6α in neurons led to brain atrophy and hippocampal degeneration in adult mice.
Moreover, double KO mice displayed accumulation of ubiquitinated proteins as well as increased levels of p-tau and Aβ1-42 in neurons of the hippocampus and cerebral cortex. Finally, double KO mice showed impaired autophagic flux and decreased v-ATPAse levels, suggesting defective autophagy and lysosomal function.

 Major concern

The contribution of UPR in maintaining neuronal proteostasis has been extensively studied in recent years (reviewed by Martínez et al., 2018, Trends Neurosci. 41(9): 610–624). The manuscript reports some interesting findings. My major concern is that the cross talk between the UPR and the autophagy lysosomal pathway in neuronal proteostasis is not sufficiently supported by data in the manuscript. Furthermore, several experimental limitations and interpretation issues need to be addressed.

Specific comments are outlined below:

- Results of qPCR and Western blot analyses from brain samples should be included in order to confirm depletion of both PERK and ATF6α in neurons of the newly generated double KO mice.

- Despite the authors assertion, a band corresponding to XBPs is detectable in single PERK or ATF6α mutant mice, albeit with some difficulty (Fig. 1A).

- With regard to the aggregation of tau, the authors report that p-tau is increased in double PERK/ATF6α KO mice compared to wild-type or single PERK/ATF6α mutants (Fig. 6A-C).  However, the difference in relative AT8 levels between double mutants and wild-type animals is marginal (Fig. 7B).

- Similarly, the authors find that Aβ peptides are accumulated in PERK/ATF6α KO double KO mice, as evidenced by immunohistochemistry data (Fig. 6D).  They should provide relevant quantifications to strengthen their results. In addition, they should also assess the levels of total and soluble Aβ1-42 peptides in the brain, as well as the levels of Aβ oligomers, since they place special emphasis on the role of perturbed UPR in disruption of neuronal proteostasis and consequent neurodegeneration.

- Given that microglial cells are essential for clearance of the Aβ extracellular plaques, the authors could more explicitly test the impact of impaired UPR on microglial activity. For example, they could perform immunohistochemistry to double stain for Aβ plaques and microglia.

- Moreover, they could test the levels of proteins implicated in synaptic function in the hippocampus of wild-type, single and double mutants so as to provide further insights into neuronal dysfunction in double PERK/ ATF6α KO mice.

- The authors show that ubiquitin-positive inclusions are increased in the hippocampus of double PERK/ATF6α mutant mice (Fig. 8A). Although this increase is clear, a Western blot showing increased levels of high-molecular- mass polyubiquitinated proteins in double KO mice compared to wild-type or single mutant animals would strengthen these results. 

- Similarly, quantification of p62 levels would further support the blockage of autophagic flux in double KO mice.

- As one of the key claims made by the authors is that UPR preserves neuronal homeostasis by regulating the autophagy-lysosome pathway, the following issues need to be further investigated:

The authors show that the protein levels of mature single chain form of cathepsin D are significantly increased in the forebrain of PERK/ATF6α double KO mice, whereas protein levels of the V0a1 subunit of v-ATPase were significantly decreased. Since the authors have already shown in a previous study (Stone et al., 2020, JCI Insight, 5(5):e132364) that cathepsin D is mis-localized in mature oligodendrocytes of PERK/ATF6α double KO mice, they could simultaneously examine cathepsin D and a lysosomal marker (for example, LAMP-1) to test for defects in localization and abundance of cathepsin in mice lacking both PERK and ATF6α in neurons.

In addition, simultaneous staining with Lysotracker that is highly selective for acidic organelles, such as lysosomes, can be indicative of lysosomal acidification and function in double PERK/ATF6α double KO mice compared to wild type and single mutant mice. To this end, the colocalization of LAMP1-GFP with LysoTracker could be examined.
Moreover, the activity of cathepsin D per se can be assessed.  
Taken together, the results of such experiments will provide significant support for the authors’ claims.

More importantly, based on data suggesting that double PERK/ATF6a KO mice show impaired lysosomal function, as well as accumulation of p-tau and Aβ peptides, the authors claim that impaired UPR disrupts neuronal proteostasis by inhibiting the autophagy lysosomal pathway. The existence of such a causal relationship should be demonstrated experimentally. For example, the authors could show whether restoration of autophagic flux and lysosome function can ameliorate the effects of perturbed UPR on neuronal proteostasis.

Textual points

Figures should consistently be commented in order to avoid confusion (for example, please see reference to Fig. 6C1-C4, Fig. 7A-C and then Fig. 6D1-D4, in the text).

Round 2

Reviewer 2 Report

In their revised manuscript, the authors have responded to most of the comments made previously. They also quantified the original immunohistochemistry data for neurons staining positive for p-Tau, Aβ accumulation and ubiquitin-positive inclusions. More specifically, the authors provide quantifications for of 4G8, CP13, PHF1, 4G8 and Aβ42 positive neurons. However, they did not assess the levels of total and soluble Aβ1-42 peptides in the brain, as well as the levels of oligomers, as requested. Nevertheless, the quantifications show that the differences between the double knockout mice and either the wild type animals or the single mutants are clear. Therefore, the authors’ hypothesis seems plausible.

Overall, the data presented in the manuscript are too preliminary to establish a causal link between the UPS and the autophagy/lysosomal pathway in the maintenance of neuronal homeostasis under physiological conditions.  However, the study now reports the possibility of such a link, as the authors have toned down their original statement. In its current form, the manuscript can therefore be accepted for publication in the IJMS.

There are a few typos that need to be corrected. For example, please correct y-axis title in Fig. 7 F, G, H and I (No. of ATg8+ neuron/0.01 mm2 etc.); it seems that an "s" is missing (it should be : No. of ATg8+ neurons/0.01 mm2 etc.).

In the legend to Figure 8 , please correct the word: sig-nificance.